

# Broad spectrum antimicrobial activities from spore-forming bacteria isolated from the Vietnam Sea

Khanh Minh Chau[1,2], Dong Van Quyen[3], Joshua M. Fraser[1], Andrew T. Smith[4], Thi Thu Hao Van[1] and Robert J. Moore[1]

[1] School of Science, RMIT University, Bundoora, Victoria, Australia
[2] NhaTrang Institute of Technology Research and Application, Vietnam Academy of Science and Technology, Nha Trang, Khanh Hoa, Vietnam
[3] Institute of Biotechnology, Vietnam Academy of Science and Technology, Cau Giay, Ha Noi, Vietnam
[4] Griffith Institute for Drug Discovery, Griffith University, Nathan, Queensland, Australia

## ABSTRACT

The widespread occurrence of pathogenic bacteria resistant to last-line antibiotics has resulted in significant challenges in human and veterinary medicine. There is an urgent need for new antimicrobial agents that can be used to control these life threating pathogens. We report the identification of antimicrobial activities, against a broad range of bacterial pathogens, from a collection of marine-derived spore-forming bacteria. Although marine environments have been previously investigated as sources of novel antibiotics, studies on such environments are still limited and there remain opportunities for further discoveries and this study has used resources derived from an under-exploited region, the Vietnam Sea. Antimicrobial activity was assessed against a panel of Gram-positive and Gram-negative bacteria, including several multi-drug resistant pathogens. From a total of 489 isolates, 16.4% had antimicrobial activity. Of 23 shortlisted isolates with the greatest antimicrobial activity, 22 were *Bacillus* spp. isolates and one was a *Paenibacillus polymyxa* isolate. Most of the antimicrobial compounds were sensitive to proteases, indicating that they were proteins rather than secondary metabolites. The study demonstrated that marine bacteria derived from the Vietnam Sea represent a rich resource, producing antimicrobial compounds with activity against a broad range of clinically relevant bacterial pathogens, including important antibiotic resistant pathogens. Several isolates were identified that have particularly broad range activities and produce antimicrobial compounds that may have value for future drug development.

Corresponding authors
Dong Van Quyen, dvquyen@ibt.ac.vn
Robert J. Moore,
rob.moore@rmit.edu.au

# INTRODUCTION

The emergence and spread of bacterial pathogens that are resistant to last–line antibiotics, for example carbapenem resistant Gram-negative pathogens, methicillin resistant *Staphylococcus aureus* (MRSA), and vancomycin resistant *Enterococci* (VRE), is of great concern in both human and veterinary medicine (*Datta & Huang, 2008*; *Loomba, Taneja & Mishra, 2010*; *Raghunath, 2010*; *O'Driscoll & Crank, 2015*; *Meletis, 2016*;

*Zaman et al., 2017*). Antibiotic resistance genes are frequently located on mobile elements such as conjugative plasmids and transposons which facilitate horizontal and vertical transmission, leading to increasing numbers of multi-resistant bacteria worldwide (*Devaud, Kayser & Bächi, 1982*; *Turner et al., 2014*). This high incidence of antibiotic resistant bacteria has serious implications for pathogen control and there is an urgent need for alternatives to the currently available antibiotics to re-control these life-threating antibiotic-resistant pathogens. The discovery of novel antibiotics from terrestrial environments over the last few decades has been challenging, due to an exhaustion of traditional antibiotic sources (*Zaman et al., 2017*). According to the FDA, approval of new medically important antibiotics had decreased by 56% over the last few decades (*Spellberg et al., 2004*) and there has been no evidence for an increase in discovery rate since that study.

The marine environment has been identified as a promising alternative source for antibiotic discovery, due to the apparently high abundance of antibiotics produced by various members of diverse marine microbial communities (*Mayer et al., 2011*; *Malve, 2016*). It has been hypothesized that marine bacteria are under unusually rigorous selection pressures because of the environmental conditions with which they must contend. They are typically exposed to low levels of nutrition and rapid changes of nutrition and physical conditions due to wave and tidal action. These harsh chemo-physical conditions have selected bacteria that deploy various mechanisms to out-compete other bacteria. Diversification of antimicrobial production is one such adaptation that can be harnessed for occupying and defending an ecological niche (*Jensen & Fenical, 1996*; *Valentine, 2007*; *Desriac et al., 2010*). It is hypothesized that marine environments may harbor novel antimicrobial producing species, which could be effective against a range of bacteria, including antibiotic resistant pathogenic bacteria. Among the diverse marine bacterial communities, members of the *Bacillus* genus, characterized as spore formers, have been shown to produce an array of structurally diverse antibiotics, including ribosomally synthesized peptides (bacteriocins), and non-ribosomal secondary metabolites such as polyketide, lipopeptide and bacilysin (*Sumi et al., 2014*). Bacteriocins are of particular interest. These small cationic peptides commonly have a narrow spectrum of activity, affecting a limited range of bacteria, and a low resistance rate, making them attractive antimicrobials for application against bacteria that have acquired resistance to the current range of available antibiotics (*Cotter, Ross & Hill, 2013*; *Mathur et al., 2017*). Many terrestrially derived bacteriocins have been characterized, but little is known of marine-derived bacteriocins.

Bacteria that produce antimicrobial compounds have the potential to be used directly, for example as probiotics, to combat pathogenic bacteria. Spore-forming bacteria have particular advantages for such applications as the spores are robust and resistant to processing and storage conditions. Hence, spore-forming antimicrobial producing bacteria such as *Bacillus* and *Paenibacillus* species can be more easily incorporated into processed food products than more vulnerable species such as *Lactobacillus* and *Lactococcus* (*Elshaghabee et al., 2017*).

Vietnam has more than 3,400 km of coastline, incorporating a variety of tropical marine ecosystems with abundant marine species and little previous study of bacteria derived

from its marine environments. These characteristics make the Vietnam Sea a promising source to explore for novel antimicrobials. The objective of this study was to test the hypothesis that bacteria that produce potentially useful antimicrobials could be found in the Vietnam Sea. A diverse collection of spore-forming bacteria was assembled and evaluated for antimicrobial molecules with activity against a range of important pathogens, including multi-antibiotic resistant pathogens.

## MATERIALS & METHODS

### Collection of marine samples and bacterial isolation

Fifty marine samples, from sponges, seaweeds, sediments, and seawaters, were collected by scuba diving at Nha Trang Bay, around Hon Mieu Island (12.191837, 109.235086), and around Hon Mot Island (12.173368, 109.271591) at depth of 5–10 m, and Hon Rua Island (12.289501, 109.242413) at depth of 1–3 m. These locations were chosen because of their known high diversity of marine habitats. The collection of environmental samples was approved by the Nha Trang Institute of Technology Research and Application.

Sponge and seaweed samples were washed three times with sterile sea water (SSW) to remove loosely attached external microbes. Ten grams of each sponge and seaweed sample were homogenized with a sterile mortar and pestle in 90 mL of sterile sea water to release microbes into the seawater. Sediments were air dried and 10 g vortexed in 90 mL of sterile seawater to detach microbes. Fifty mL of sea waters were centrifuged at 2,500 g for 10 minutes and the 1 mL at the bottom of the tube was used to resuspend pelleted material. Aliquots of all samples were heated at 80 °C for 20 minutes to kill non-spore-forming bacteria, followed by serial dilution for plating and isolation. From each dilution 100 µL was plated onto laboratory-prepared marine agar (LPMA) (2.5 g/L yeast extract, 5.0 g/L peptone, 1.0 g/L dextrose, 0.2 g/L $K_2HPO_4$, 0.05 g/L $MgSO_4.7H_2O$, 750 mL/L aged sea water, 250 mL/L tap water, pH = 7.5). The LPMA was Youshimizu and Kimura medium as modified by *Mikhailov, Romanenko & Ivanova (2006)*. Aged sea water was prepared by storing fresh sea water in the dark for 2–4 weeks to stabilize or neutralize the heavy metals or toxic compounds which may affect bacterial recovery. The inoculated plates were incubated at 30 °C for up to 4 days. Colonies were re-streaked until only colonies with similar morphologies were observed on the plates. Pure colonies were scrapped off plates to mix in laboratory-prepared marine broth (LPMB) supplemented with 20% glycerol and stored at −80 °C.

### Primary screening for antimicrobial activity by cross-streak assay

Each bacterial isolate was streaked vertically onto an LPMA plate and incubated overnight at 30 °C. The six indicator strains used in the initial screening were, *S. aureus, B. cereus, C. albicans, E. coli, P. aeruginosa,* and a methicillin resistant *S. aureus.* Each indicator strain was streaked, horizontally, from the edge of the plate to the pre-grown isolate streak. The cross-streak assay plates were then incubated in the growth conditions appropriate for each indicator strain, as detailed in Table 1. The gap between indicator strain and the test isolates' growth indicated the presence or absence of antimicrobial activity. After inspection of the strength and spectrum of antimicrobial activity, a short-list of isolates with the strongest

**Table 1** List of indicators strains used for antimicrobial screening experiments.

| No | Strain | Origin/Strain storage | Media/growth condition |
|---|---|---|---|
| | **Gram-positive bacteria** | | |
| 1 | *Streptococcus faecalis* | ATCC 29212 | MH/ 37 °C/aerobic |
| 2 | *Lactobacillus plantarum* | RMIT university | MRS/37 °C/aerobic |
| 3 | *Bacillus cereus* | ATCC 10876 | MH/ 30 °C/aerobic |
| 4 | *Staphylococcus aureus* | ATCC25923 | MH/37 °C/aerobic |
| 5 | *Listeria monocytogenes* | Human pathogen, RMIT | BA /37 °C/aerobic |
| 6 | *Clostridium perfringens* | Chicken pathogen, RMIT | MH /37 °C/anaerobic |
| | **Gram- negative bacteria** | | |
| 7 | *Salmonella* Enteritidis | ATCC 13076 | MH/ 37 °C/aerobic |
| 8 | *Escherichia coli* | ATCC 25922 | MH/ 37 °C/aerobic |
| 9 | *Pseudomonas aeruginosa* | ATCC 15442 | MH/ 37 °C/aerobic |
| 10 | *Campylobacter jejuni* | Chicken pathogen, RMIT | BA/37 °C/microaerophilic |
| 11 | *Campylobacter coli* | Chicken pathogen, RMIT | BA/37 °C/ microaerophilic |
| | **Yeast** | | |
| 12 | *Candida albicans* | ATCC 10231 | MH/30 °C/aerobic |
| | **Antibiotic resistant pathogens** | | |
| 13 | Methicillin resistant *Staphylococcus aureus* (MRSA) | Human pathogen, RMIT | MH/37 °C/aerobic |
| 14 | Vancomycin resistant *Enterococcus faecalis* (VRE) | Human pathogen, RMIT | MRS/37 °C/aerobic |
| 15 | Multidrug resistant *Klebsiella pneumonia* (MRKP) | Human pathogen, RMIT | MH/37 °C/aerobic |

**Notes.**

Standard media were MRS, De Man, Rogosa and Sharpeagar; MH, Muller Hilton agar; BA, Muller Hilton agar supplemented with 5% sheep blood. Microaerophilic and anaerobic condition were obtained with Gas-Pak.

antimicrobial activity was chosen for more detailed analysis. Their antagonistic activities against 14 pathogens were evaluated.

The cross-streak assay was also used to detect antimicrobial activities produced by the 23 short-listed isolates against the other members of the group. Each marine isolate was streaked down the middle of LPMA plates, incubated overnight at 30 °C to allow growth and production of antimicrobial compounds, and then the other 22 isolates were streaked from the edge of the plate to the central streak. The plate was incubated overnight at 37 °C and antimicrobial activity was determined based on the size of the clear zone between test and indicator bacterial streaks.

## Well-diffusion assay

Well-diffusion assays were used to determine if antimicrobial activity was secreted into liquid culture supernatant. Muller Hilton agar plates were swabbed with a suspension of indicator bacteria ($OD_{600}$∼0.08–0.1). Wells (6 mm diameters) were punched from the agar using a sterilized cork cutter and then 50 μL of cell free supernatant (CFS) from test cultures grown for 24 hours was added to a well. The CFS in the wells was air dried and plates were incubated at the optimal conditions for growth of the indicator strain (Table 1). The inhibitory effects of antimicrobial within the CFS were observed by appearance of a zone of clearing of the indicator bacteria around the well.

### 16S rRNA gene amplification, sequencing, and phylogenetic analysis

Bacteria were sub-cultured into 5 mL of LB from a single colony. Total genomic DNA (gDNA) was extracted from the overnight culture using a guanidine thiocyanate method, as previously describes (*Pitcher, Saunders & Owen, 1989*), and then used as template to amplify the 16S rRNA gene sequences. PCR was conducted using primers with the sequences (5′–3′) GGCGTGCCTAATACATGCAA and TACAAGGCCCGGGAACGT. The primers were designed based on the alignment of the 16S rDNA sequences of *Bacillus* isolates. PCR condition comprised initial denaturation at 98 °C for 30 seconds, 30 cycles of 98 °C for 5 seconds, 56 °C for 10 seconds and 72 °C for 20 seconds, extension at 72 °C for 2 minutes; and 4 °C for 10 minutes. PCR products were checked by electrophoresis in 1% agarose gel, subsequently purified by QIAquick PCR Purification Kit (Qiagen), and then Sanger-sequenced (Micromon, Monash University, Australia). The 16S rRNA gene sequences are deposited under NCBI GenBank accession numbers MT758446–MT758468. The raw reads were trimmed of unclear nucleotides at both 5′ and 3′ terminal ends, and subsequently blastn was used to search for homologies in the Bacterial 16S rDNA Database (https://blast.ncbi.nlm.nih.gov/Blast.cgi). The highly homologous 16S rDNA sequences were download for phylogenetic tree construction. All the sequences were aligned using ClustalW (*Thompson, Higgins & Gibson, 1994*), and the phylogenetic tree was subsequently constructed in MEGA7 using the neighbor joining method with bootstrap tests performed 1000 times and pairwise detection (*Kumar, Stecher & Tamura, 2016*).

### Sensitivity of antimicrobial activities to enzyme and heat treatments

Enzymatic treatments of CFSs were conducted for 3 hours at 37 °C with pronase-E from *Streptomyces griseus*; proteinase K; trypsin; and lipase, at final concentrations of 2 mg/mL. All enzymes were purchased from Sigma Aldrich. Heat stability of antimicrobials was determined by incubation of CFSs at 60 °C, 80 °C, and 100 °C for 30 minutes, 60 minutes, and 3 hours. The antimicrobial activities of the treated CFS preparation were evaluated by well-diffusion assay, against *Clostridium perfringens*.

### Growth properties, antibiotic susceptibility testing, and enzyme production of isolates

The ability of the short-listed isolates to grow on different media was evaluated by spotting 5μL of bacterial cultures onto several different media, including low nutrition media such as marine agar (BD Difco 2216) and LPMA, and rich nutritious media such as LB agar and Muller Hilton agar, and incubated at 30 °C overnight. Aliquots of cultures were spotted on LB agar plates, and incubated under microaerophilic, aerobic, and anaerobic conditions at 30 °C, and aerobically incubated at 40 °C and 50 °C. To measure pH tolerance, bacterial cultures were spotted onto LB plates in which the media had been adjusted to pH 5.0, 6.0, 7.0, 8.0, and 9.0. Sodium chloride tolerance was determined on LB plates supplemented with 0%, 1%, 2%, 4%, 6%, 8%, 10%, 12% and 15% (w/v) NaCl; while bile salt tolerance was carried out on the LB plates supplemented bile salt at final concentration of 0.1M, 0.2M, 0.3M, 0.4M, 0.5M, 0.6M, 0.7M respectively (Bile Salts Mixture No. 3; Neogen Corporation). For antibiotic susceptibility testing, LB agar plates were supplemented

**Table 2  Antimicrobial producing bacteria identified from marine samples.**

|  | Number of marine samples collected | Number of isolated spore-forming bacteria | Number of antimicrobial isolates | Percentage of isolates with antimicrobial activity |
|---|---|---|---|---|
| Sponges | 16 | 183 | 28 | 14.8% |
| Seaweeds | 13 | 92 | 15 | 16.3% |
| Sediments | 13 | 81 | 16 | 19.8% |
| Sea water | 8 | 33 | 6 | 18.2% |
| Total | 50 | 389 | 64 | 16.5% |

with tetracycline, ampicillin, nalidixic acid, and kanamycin at final concentration of 50 µg/mL. The production of proteases and cellulases, amylase was evaluated respectively by spotting of 5 µL of bacterial cultures onto skim milk agar; carbon deficient media (CDM) supplemented with 1% carboxymethylcellulose (CMC), and CDM supplemented with 1% soluble starch. CDM contained 0.1 g/L yeast extract; 0.5 g/L peptone; 16.0 g/L agar. Plates were incubated at 30 °C overnight, followed by flushing the CMC plates and starch agar plates with Gram's iodine solution (Sigma Aldrich) for 1 minute. Positive reactions, indicating enzymatic activity, were noted via halo zones around the bacteria spots.

## Statistical analysis

A two-tailed $t$-test was used to assess whether there were any statistically significant differences in the rate of isolation of antimicrobial producing isolates from the different marine sources. The statistical analysis was computed in Microsoft Excel with a 5% level of probability used to indicate significance.

# RESULTS

## Antimicrobial activity of thermally resistant, marine spore-forming bacteria

A total of 389 heat resistant, spore-forming, bacterial isolates were cultured from 50 marine samples (Table 2). They demonstrated a range of colony morphologies (Fig. 1). Primary screening against six indicator strains, using the cross-streak assay, identified 64 isolates (16.4%) with antimicrobial activity (Figs. 2A and 2C). The proportion of isolates that exhibited activity against Gram-positive indicator strains, *Bacillus cereus* (93.7%), *Staphylococcus aureus* (84.3%), *Streptococcus faecalis* (87.5%), was higher than the proportion that exhibited activity against the Gram-negative indicators, *Escherichia coli* (50%) and *Pseudomonas aeruginosa* (4.6%), and the yeast, *Candida albicans* (21.4%). A two-tailed $t$-test showed that the there was no statistically significant difference in the rate of isolation of antimicrobial producing bacteria from the different types of marine samples. The p values for the pairwise comparisons amongst all the sample types ranged from 0.37 to 0.77.

The spectra of antimicrobial activities, exhibited by a select group of the 23 most potent isolates from the primary screen, were determined against an expanded panel of 14 indicator strains, including important multidrug resistant pathogens. The analyses were performed using two assays: a cross-streak assay and a well-diffusion assay. There

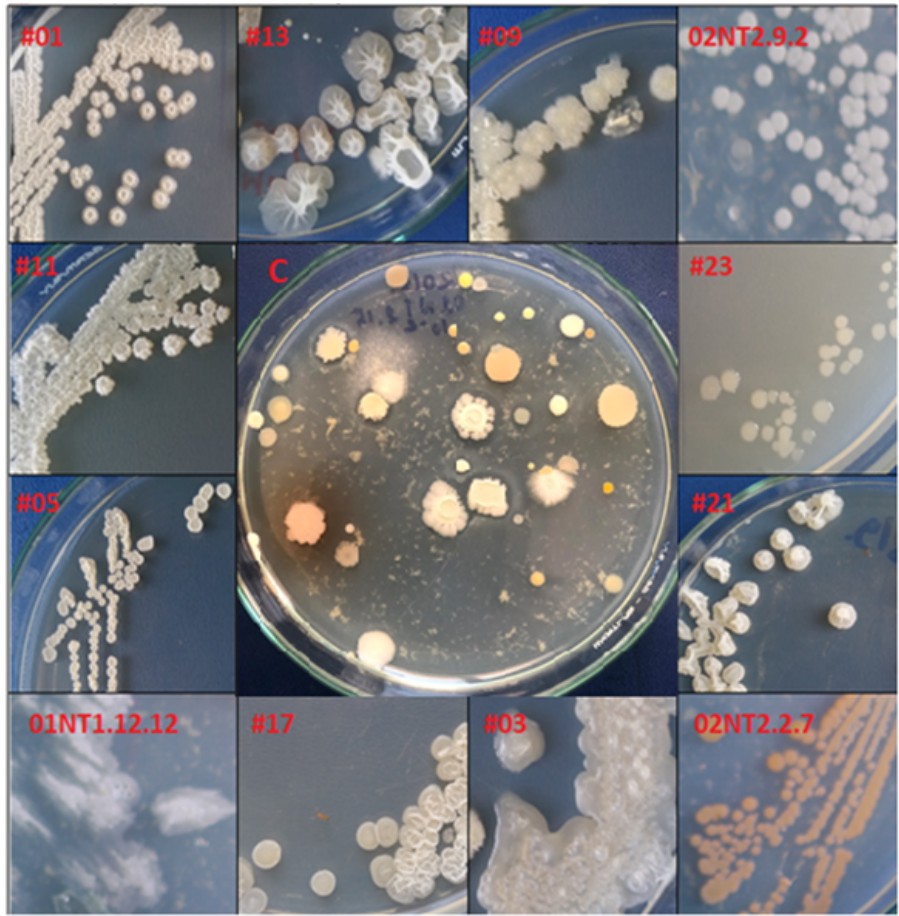

**Figure 1 Morphological diversity of the marine isolates.** Colony morphologies of some pure marine spore-forming bacteria. The symbol # indicated the isolate number amongst the 23 short-listed isolates. 'C' is an example of a primary culture plate, in this case from a seaweed sample, from which isolates were subsequently colony purified. Photo credit: Khanh Minh Chau.

was considerable variation in the strength and spectra of antimicrobial activity across the 23 short-listed isolates. The pathogenic indicator strains most commonly affected by the antimicrobial compounds expressed by the screened isolates were the Gram-positive species (Table 3; Fig. 3). Of the bacterial indicators, *Clostridium perfringens, B. cereus* and *S. aureus* were inhibited by 83% (19/23) of the test isolates, whereas the proportion with activity against the Gram-negative indicators; *Campylobacter jejuni,* 70% (16/23), *Campylobacter coli,* 61% (14/23), was lower. Two antibiotic resistant Gram-positive pathogens; MRSA and VRE were inhibited by respectively 83% (19/23) and 57% (13/23) of isolates, while an antibiotic resistant Gram-negative pathogen; multidrug resistant *Klebsiella pneumonia* (MRKP), was inhibited by only one isolate, *P. polymyxa* #23. The Gram-negative bacteria *P. aeruginosa* was also inhibited by *P. polymyxa* #23. The growth of foodborne pathogens; *Listeria monocytogenes, E. coli* and *Salmonella* Enteriditis were depressed by, respectively,

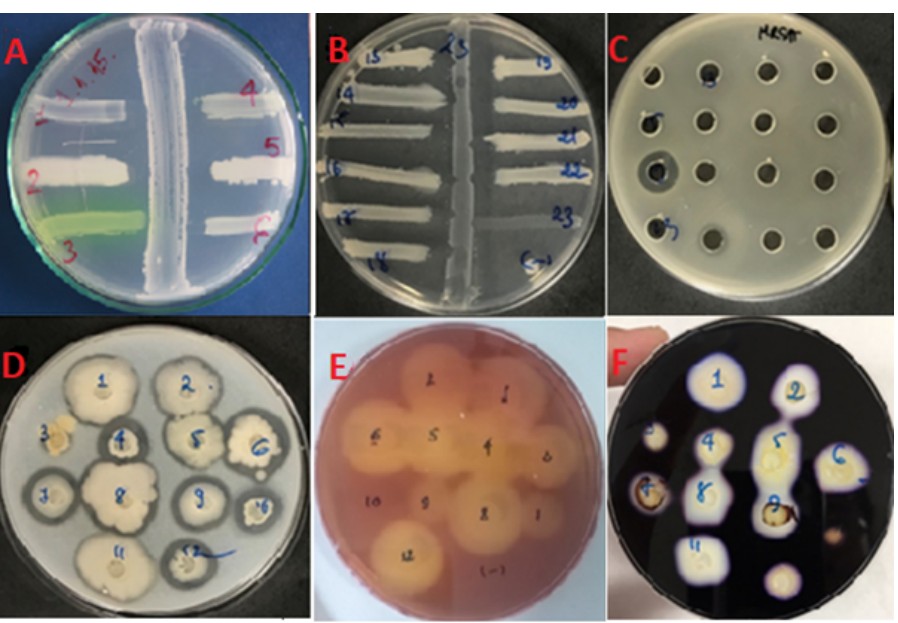

**Figure 2** **Bioactivity assays.** Antimicrobials activity screened by (A, B) cross-streak assay and (C) well-diffusion assay. (A) The cross-streak assay to identify antimicrobial producing bacteria against six indicators including; (1) *S. faecalis*; (2) *B. cereus*; (3) *P. aeruginosa*; (4) *E. coli*; (5) *S. aureus*; (6) *C. albicans*. (B) The illustration of cross-activity exhibited by *P. polymyxa* #23 (as a representative) against growth of other marine species. Notably, *P. polymyxa* killed all other marine *Bacillus* excepting itself #23 (as negative control). (C) The growth of MRSA as indictor was depressed by two isolates' culture supernatants. (D) Proteolytic activity screened on skim milk agar. (E) Cellulose degradation activity screened on CMC agar. (F) Amylase production screened on starch agar. Photo credit: Khanh Minh Chau.

78%, 44%, and 57% of the isolates. None of the selected isolates had inhibitory activity against *C. albicans*.

The cross-streak assay was found to be more sensitive than the well-diffusion assay. In most cases antimicrobial activity was detected in both assays but occasionally the activity was less or absent in the well-diffusion assay. This difference was particularly apparent when *S. aureus,* MRSA and most of Gram-negative pathogens, *C. jejuni, C. coli, E. coli* and *S.* Enteriditis, were used as the indicator strains.

## Taxonomic analysis of antimicrobial isolates

Of the 23 of the isolates selected as expressing the most antimicrobial activity, there were 22 *Bacillus* species isolates and 1 *Paenibacillus* isolate. Their 16S rRNA gene sequences shared 99%–100% identity with, mostly, terrestrially derived species (Table 4). The phylogenetic tree demonstrated the relationship amongst the marine isolates and to terrestrial isolates (Fig. 4). The most commonly identified species amongst the short-listed isolates were *B. subtilis* (10 isolates) and other members of *B. subtilis* group such as *B. amyloliquefaciens* (5 isolates), *B. licheniformis* (1 isolate), and *B. safensis* (1 isolate). In addition, members of other *Bacillus* groups were also identified including *B. pacificus* (2 isolates), belonging to *B. cereus* group; *B. halotolerans* (3 isolates), and *P. polymyxa* (1 isolate)

Chau et al. (2020), *PeerJ*, DOI 10.7717/peerj.10117

**Table 3** **Antimicrobial activity of short-listed isolates against 14 indicators strains.** The antimicrobial activities were evaluated by well diffusion assay (WD) and cross streak assay (CS).

| | #01 | | #02 | | #03 | | #04 | | #05 | | #06 | | #07 | | #08 | | #09 | | #10 | | #11 | | #12 | |
|---|---|---|---|---|---|---|---|---|---|---|---|---|---|---|---|---|---|---|---|---|---|---|---|---|
| | CS | WD | CS | WD | CS | WD | CS | WD | CS | WD | CS | WD | CS | WD | CS | WD | CS | WD | CS | WD | CS | WD | CS | WD |
| *Clostridium perfringens* | ++ | ++ | ++ | + | – | – | + | + | ++ | +++ | ++ | ++ | – | – | +++ | ++ | – | – | + | + | +++ | ++ | ++ | + |
| *Staphylococcus aureus* | +++ | ++ | ++ | – | + | – | – | – | + | + | +++ | – | – | – | +++ | – | – | – | ++ | ++ | ++ | – | ++ | – |
| *Bacillus cereus* | ++ | ++ | ++ | ++ | – | – | + | – | + | ++ | +++ | ++ | – | – | +++ | +++ | ++ | – | + | + | +++ | ++ | + | + |
| *Listeria monocytogenes* | ++ | ++ | ++ | + | – | – | + | – | – | – | +++ | ++ | + | – | ++ | + | – | – | – | – | ++ | + | – | – |
| *Lactobacillus plantarum* | ++ | + | + | + | + | + | – | – | – | – | ++ | ++ | – | – | ++ | ++ | – | – | + | + | ++ | ++ | – | – |
| *Candida albicans* | – | – | – | – | – | – | – | – | – | – | – | – | – | – | – | – | – | – | – | – | – | – | – | – |
| *Escherichia coli* | + | – | + | – | – | – | – | – | + | – | – | – | – | – | + | + | – | – | – | – | + | + | + | – |
| *Salmonella* Enteritidis | + | – | – | – | – | – | – | – | – | – | ++ | – | + | – | ++ | + | – | – | – | – | ++ | + | + | – |
| *Campylobacter jejuni* | ++ | + | ++ | + | – | – | – | – | ++ | – | ++ | ++ | + | – | ++ | ++ | – | – | – | – | +++ | ++ | – | – |
| *Campylobacter coli* | + | +++ | ++ | ++ | – | – | ++ | +++ | + | – | + | +++ | + | – | + | +++ | – | – | – | – | ++ | +++ | ++ | ++ |
| *Pseudomonas aeruginosa* | – | – | – | – | – | – | + | – | – | – | – | – | – | – | – | – | – | – | – | – | – | – | – | – |
| MRSA | +++ | ++ | ++ | – | + | – | – | – | + | + | +++ | – | – | – | +++ | – | – | – | ++ | ++ | ++ | – | ++ | – |
| VRE | ++ | ++ | ++ | ++ | – | – | + | + | – | – | + | – | – | – | +++ | +++ | – | – | ++ | ++ | ++ | ++ | + | ++ |
| MRKP | – | | – | | – | – | – | – | – | – | – | – | – | – | – | – | – | – | – | – | – | – | – | – |

| | #13 | | #14 | | #15 | | #16 | | #17 | | #18 | | #19 | | #20 | | #21 | | #22 | | #23 | | Percent (%) | |
|---|---|---|---|---|---|---|---|---|---|---|---|---|---|---|---|---|---|---|---|---|---|---|---|---|
| | CS | WD | CS | WD | CS | WD | CS | WD | CS | WD | CS | WD | CS | WD | CS | WD | CS | WD | CS | WD | CS | WD | CS | WD |
| *Clostridium perfringens* | +++ | +++ | ++ | + | ++ | + | ++ | + | ++ | ++ | ++ | + | ++ | ++ | ++ | + | +++ | +++ | ++ | – | ++ | + | 86.7 | 82.6 |
| *Staphylococcus aureus* | +++ | – | +++ | – | +++ | – | +++ | – | ++ | – | ++ | – | ++ | +++ | +++ | – | – | – | +++ | – | +++ | + | 82.6 | 21.7 |
| *Bacillus cereus* | ++ | +++ | +++ | ++ | +++ | ++ | +++ | ++ | ++ | ++ | ++ | ++ | ++ | ++ | ++ | ++ | ++ | ++ | ++ | ++ | ++ | ++ | 82.6 | 82.6 |
| *Listeria monocytogenes* | ++ | ++ | +++ | +++ | +++ | ++ | +++ | ++ | ++ | ++ | ++ | +++ | ++ | – | +++ | +++ | ++ | ++ | ++ | ++ | ++ | – | 78.2 | 60.8 |
| *Lactobacillus plantarum* | ++ | +++ | – | – | – | – | – | – | – | – | – | – | ++ | ++ | – | – | ++ | ++ | – | – | + | + | 47.8 | 47.8 |
| *Candida albicans* | – | – | – | – | – | – | – | – | – | – | – | – | – | – | – | – | – | – | – | – | – | – | 0 | 0 |
| *Escherichia coli* | ++ | + | – | – | – | – | – | – | – | – | – | – | ++ | – | ++ | – | – | – | – | – | ++ | ++ | 43.5 | 17.3 |
| *Salmonella* Enteritidis | ++ | – | – | – | + | – | + | – | + | – | + | – | ++ | – | ++ | – | +++ | + | + | – | +++ | ++ | 56.5 | 17.3 |
| *Campylobacter jejuni* | + | – | ++ | – | + | – | + | – | ++ | – | ++ | – | – | – | – | – | + | – | ++ | – | ++ | ++ | 69.5 | 26 |
| *Campylobacter coli* | + | ++++ | – | – | – | – | – | – | ++ | – | + | – | ++ | – | + | – | ++ | ++++ | + | – | +++ | ++ | 60.8 | 43.5 |
| *Pseudomonas aeruginosa* | – | – | – | – | – | – | – | – | – | – | – | – | – | – | – | – | – | – | – | – | ++ | ++ | 4.3 | 4.3 |
| MRSA | +++ | – | +++ | ++ | +++ | ++ | +++ | ++ | ++ | ++ | ++ | +++ | ++ | + | +++ | ++ | – | – | +++ | ++ | +++ | ++ | 82.6 | 21.7 |
| VRE | ++ | +++ | – | – | – | – | – | – | + | + | + | + | + | ++ | – | – | ++ | ++ | – | – | + | + | 56.5 | 52.2 |
| MRKP | – | – | – | – | – | – | – | – | – | – | – | – | – | – | – | – | – | – | – | – | ++ | ++ | 4.3 | 4.3 |

**Notes.**

[+] zone of inhibition observed with clear halo of growth inhibition in at least one time point

[++ and +++] increased activity as assessed visually by the increased diameter of the inhibition zone

[-] no inhibition

The percentage values were calculated based on the ratio between number of isolates with antimicrobial activity against the indicator strain and the total 23 isolates tested.

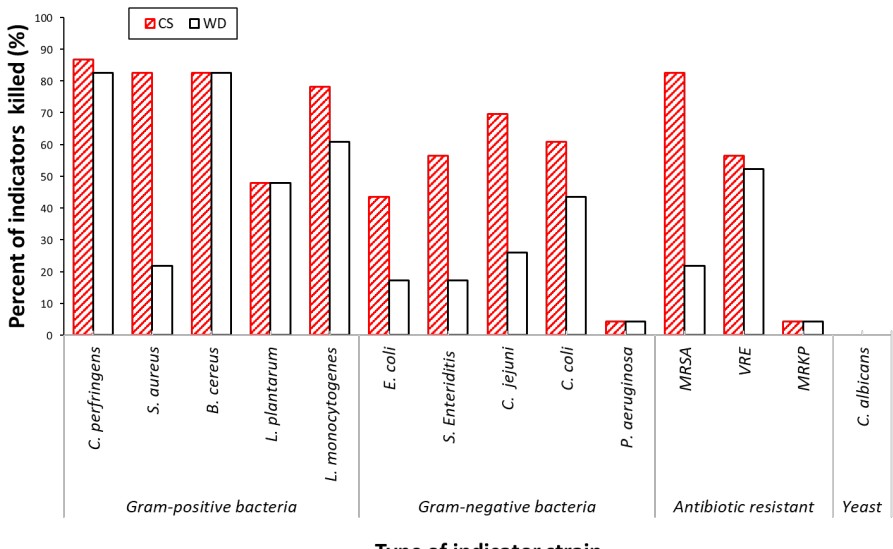

**Figure 3** **Antimicrobial activities amongst the short-listed 23 isolates against 14 pathogenic indicator strains.** Chart was constructed based on the value obtained from Table 5. WD, Well-diffusion assay; CS, Cross-streak assay.

## Many of the antimicrobials are proteinaceous compounds

It has previously been shown that protein and non-protein antimicrobial compounds are produced by some terrestrial *Bacillus* isolates. The proteinaceous nature of some of the antimicrobial activities identified in the marine *Bacillus* isolate collection was demonstrated by their susceptibility to protease action. For these assays *C. perfringens* was selected as the indicator strain because amongst the indictor strains tested (Table 3) it displayed sensitivity to the highest percentage of the *Bacillus* produced antimicrobial compounds. The protease sensitivity of the anti-*C. perfringens* activity of the isolates was determined by digesting cell culture supernatants used in the well-diffusion assay with proteases. In 16 of the 19 isolates tested the antimicrobial activities against *C. perfringens* were reduced or lost after treatment with at least one proteolytic enzyme (Table 5). The pronase-E enzyme completely eliminated anti-*C. perfringens* activity from 10 isolates, while proteinase-K removed the activity from 4 isolates. Some antimicrobials (produced by isolates #06, #11, and #21) were affected by both proteinases and a lipase. Other isolates showed reduced, but not eliminated, antimicrobial activity following protease treatment (#01, #08, #20, # 21). The antimicrobial activities produced by isolates #5, #18, #19 were not affected by any of the enzymes used. Heat treatment at 60 °C for 30 minutes abolished the activity of 12/19 antimicrobials while 7/19 retained activity. Of the isolates in which antimicrobial activity was not affected by proteases, isolates #05 and #19 were also resistant to heat treatment whereas anti-*C. perfringens* activity was abolished when #18 supernatant was heated.

## Antimicrobial activity between the short-listed isolates

It is hypothesised that bacteria produce antimicrobial compounds to compete within an ecological niche. We therefore investigated whether this group of isolates exhibited any
**Table 4  Closest species, by 16S rRNA gene similarity, of antimicrobial producing isolates.** Identified by BLASTn of 16S rDNA sequence against the NCBI 16S rDNA database.

| Isolates | Accession[*] | Closest species | Identity (%) |
|---|---|---|---|
| #01 | MT758446 | *B. halotolerans/B. mojavensis/B. subtilis subsp. spizizenii* | 100 |
| #02 | MT758447 | *B. subtilis/B. tequilensis* | 100 |
| #03 | MT758448 | *B. licheniformis/B. haynesii* | 99.92 |
| #04 | MT758449 | *B. subtilis/B. tequilensis* | 100 |
| #05 | MT758450 | *B. subtilis/B. tequilensis* | 100 |
| #06 | MT758451 | *B. amyloliquefaciens* | 99.92 |
| #07 | MT758452 | *B. pacificus/B. paranthracis/B. cereus* | 100 |
| #08 | MT758453 | *B. amyloliquefaciens* | 99.92 |
| #09 | MT758454 | *B. pacificus/B. paranthracis/B. cereus* | 100 |
| #10 | MT758455 | *B. safensis/B. australimaris/B. pumilus* | 100 |
| #11 | MT758456 | *B. amyloliquefaciens* | 99.92 |
| #12 | MT758457 | *B. subtilis/B. tequilensis* | 100 |
| #13 | MT758458 | *B. amyloliquefaciens* | 99.92 |
| #14 | MT758459 | *B. subtilis/B. tequilensis* | 100 |
| #15 | MT758460 | *B. amyloliquefaciens* | 99.92 |
| #16 | MT758461 | *B. subtilis/B. tequilensis* | 99.87 |
| #17 | MT758462 | *B. subtilis/B. tequilensis* | 99.92 |
| #18 | MT758463 | *B. subtilis/B. tequilensis* | 99.76 |
| #19 | MT758464 | *B. halotolerans/B. mojavensis* | 100 |
| #20 | MT758465 | *B. subtilis/B. tequilensis* | 100 |
| #21 | MT758466 | *B. halotolerans/B. mojavensis* | 99.84 |
| #22 | MT758467 | *B. subtilis/B. tequilensis* | 100 |
| #23 | MT758468 | *Paenibacillus polymyxa* | 99.45 |

**Notes.**
[*]NCBI GenBank accession number

antimicrobial activities against each other. Growth inhibition was detected, using the cross-streak assay, between strains of the same species for isolates *B. amyloliquefaciens* #06, #08, and *B. subtilis* #05. Cross-species antimicrobial activities were detected for many of the isolates, including, *B. amyloliquefaciens* #06, #08, #11, #13, *B. licheniformis* #03; *B. safensis* #10; *B. halotolerans* #01, and *P polymyxa* #23. Particularly, *P. polymyxa* #23 depressed growth of all the marine *Bacillus* assayed (Fig. 2B, Table 6)

## Growth characteristics of antimicrobial-producing isolates

The basic growth parameters of the 23 isolates were qualitatively evaluated (Table 7). All isolates could grow on various kinds of media. Growth was better in rich nutritious medias such as Luria-Bertani (LB), brain heart infusion (BHI), Müller Hinton agar (MH), and blood agar (BA), than growth in less nutritious media such as lab-prepared marine agar (LPMA) and marine broth (MB). All 23 isolates grew vigorously under both aerobic and microaerophilic conditions but not in anaerobic condition, under incubations temperatures of 30 °C, 37 °C, 40 °C, 50 °C, and under pHs ranging from 6.0 to 9.0. Some isolates had reduced growth at pH5.0. All isolates could grow in media supplemented with 5% NaCl, and all but isolate #23 grew in 7% NaCl. No isolate could tolerate 12% NaCl. The marine

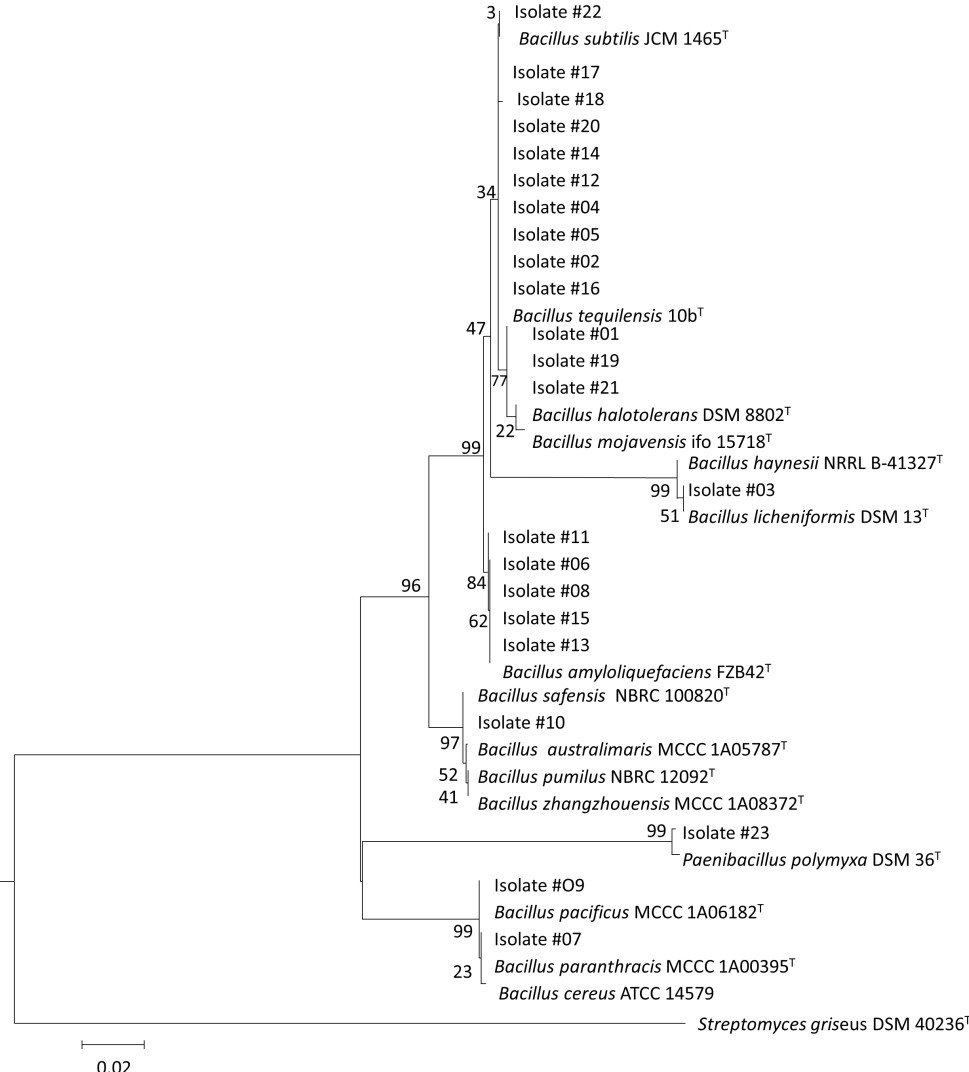

**Figure 4  Phylogenetic tree of the short-listed 23 antimicrobial producing isolates.** The neighbor-joining phylogenetic tree was constructed using the maximum composite likelihood method, bootstrap method of 1,000 replication and pairwise deletion by MEGA 7. In this tree, 23 marine isolates were displayed by number value, while all reference strains bacteria were closely related bacteria identified and downloaded from NCBI.

environment, from which the isolates were derived, typically has a salt content of 3.5%. No isolates grew in LB media supplemented with bile salts, even at lowest concentration tested (0.1M). All isolates were sensitive to 3 antibiotics; nalidixic acid (50 µg/mL), kanamycin (50 µg/mL) and tetracycline (50 µg/mL), but 14 of the 23 isolates were resistant to ampicillin (50 µg/mL). All the isolates had detectable levels of cellulolytic, proteolytic or amylolytic activity (Figs. 2D–2F).

**Table 5   Enzymatic sensitivity and heat stability profile of antimicrobial activities.** These 19 isolates showed antagonistic activities against *Clostridium perfringens*.

| Isolate | Activity* | Proteinase K (proK) | Pronase E (proE) | Trypsin | Lipase | Sensitive to enzymes | Resistant to (t °C) |
|---|---|---|---|---|---|---|---|
| #01 | 13.67 ± 0.58 | 10.67 ± 0.58 | 10.67 ± 0.58 | 14.00 | 13.67 ± 0.58 | proE, proK | 60 |
| #02 | 11.00 | 9.00 | – | 10.67 ± 0.58 | 11.00 | proE, proK | 60 |
| #04 | 9.00 | – | – | – | 9.00 | proE, proK, trypsin | – |
| #05 | 23.67 ± 0.58 | 23.67 ± 0.58 | 23.67 ± 0.58 | 23.67 ± 0.58 | 23.67 ± 0.58 | – | 60 |
| #06 | 14.67 ± 0.58 | 13.00 ± 1.00 | 10.33 ± 0.58 | – | – | proE, trypsin, lipase | – |
| #08 | 13.67 ± 0.58 | 12.00 | 11.33 ± 0.33 | 13.33 ± 0.33 | 14.00 | proE, proK | 60 |
| #10 | 9.00 | – | – | – | 9.00 | proE, proK, trypsin | – |
| #11 | 17.00 | 15.00 | – | 15 ± 0.58 | – | proE, proK, lipase | 60 |
| #12 | 11.67 ± 0.58 | 8.67 ± 0.58 | – | – | 12.00 | proE, proK, trypsin | – |
| #13 | 16.00 | 16.00 | 16.00 | 12.57 ± 0.58 | 18.33 ± 0.58 | trypsin | 60 |
| #14 | 8.67 ± 0.58 | 8.67 ± 0.58 | – | – | – | proE, proK, trypsin, lipase | – |
| #15 | 8.67 ± 0.58 | – | – | 9.00 | 9.00 | proE, trypsin | – |
| #16 | 12.00 ± 0.58 | – | – | – | 12.00 | proE, proK, trypsin | – |
| #17 | 12.33 ± 0.58 | 9.67 ± 0.58 | – | – | 8.67 ± 0.58 | proE, proK, trypsin | – |
| #18 | 9.67 ± 0.58 | 10.33 ± 0.58 | 10.00 | 10.33 ± 0.58 | 10.00 | – | – |
| #19 | 13.67 ± 0.58 | 14.00 | 13.67 ± 0.58 | 13.67 ± 0.58 | 12.67 ± 0.58 | – | 60 |
| #20 | 11.00 | 11.00 | 8.67 ± 0.58 | 11.00 | 11.00 | proE | – |
| #21 | 15.00 | 11.00 | 12.33 ± 0.58 | 15.00 | 11.33 ± 0.58 | proE, proK, lipase | – |
| #23 | 13.00 | 12.33 ± 0.58 | – | 12.33 ± 0.58 | 12.67 ± 0.58 | proE | – |

**Notes.**
*Value indicated the diameter of halo zone of antimicrobial activity in millimetres including well diameter of 6 mm

'–'indicates no zone of clearing.

Underlined enzyme indicated a loss of activity after enzymatic treatment. The values of temperature resistance were recorded after 3 hours of incubation. The numbers in the table represented the mean value (±standard deviation) of killing diameters from the measurement of three replicates. These 19 isolates showed antagonistic activities against *Clostridium perfringens*.

## DISCUSSION

There is growing interest in the marine environment as a potential source of bacteria that produce novel antimicrobial compounds, particularly bacteriocins. These antimicrobial peptides have been gaining interest as potential drug candidates for clinical treatment of antibiotic resistant pathogens (*Cotter, Ross & Hill, 2013*). In this study, spore-forming bacteria isolated from the coastal marine environment of Nha Trang (Vietnam Sea) were screened to identify *Bacillus* isolates that produce antimicrobial compounds. Members of the *Bacillus* genus were targeted because they are well-recognized as producers of

Chau et al. (2020), *PeerJ*, DOI 10.7717/peerj.10117

Peerj

**Table 6** **Antimicrobial activities amongst the short-listed isolates.** The analysis was conducted using the cross-streak method.

| Isolate | #02 | #04 | #05 | #12 | #14 | #16 | #17 | #18 | #20 | #22 | #06 | #08 | #11 | #13 | #15 | #07 | #09 | #03 | #10 | #01 | #19 | #21 | #23 |
|---|---|---|---|---|---|---|---|---|---|---|---|---|---|---|---|---|---|---|---|---|---|---|---|
| #02 *B. subtilis* | – | – | – | – | – | – | – | – | – | – | – | + | – | – | – | – | + | – | – | – | – | – | – |
| #04 *B. subtilis* | – | – | – | – | – | – | – | – | – | – | – | + | – | – | – | – | – | – | – | – | – | – | – |
| #05 *B. subtilis* | + | + | – | + | – | + | + | + | + | + | + | + | + | + | + | + | + | + | + | – | – | + | + |
| #12 *B. subtilis* | – | – | – | – | – | – | – | – | – | – | – | – | + | – | – | – | – | – | – | – | – | – | – |
| #14 *B. subtilis* | – | – | – | – | – | – | – | – | – | – | – | – | – | – | – | – | – | – | – | – | – | – | – |
| #16 *B. subtilis* | – | – | – | – | – | – | – | – | – | – | + | + | – | + | – | – | – | – | ++ | + | + | ++ | + |
| #17 *B. subtilis* | – | – | – | – | – | – | – | – | – | – | + | + | – | ++ | – | – | – | – | ++ | – | ++ | ++ | + |
| #18 *B. subtilis* | – | – | – | – | – | – | – | – | – | – | + | + | – | ++ | – | – | + | – | + | – | ++ | ++ | ++ |
| #20 *B. subtilis* | – | – | + | – | – | – | – | – | – | – | – | – | – | ++ | – | – | – | – | – | – | + | + | – |
| #22 *B. subtilis* | – | – | – | – | – | – | – | – | – | – | – | – | – | ++ | – | – | + | – | – | – | ++ | ++ | ++ |
| #06 *B. amyloliquefaciens* | + | + | + | + | – | ++ | ++ | ++ | ++ | ++ | – | + | + | + | – | + | + | + | + | + | ++ | ++ | ++ |
| #08 *B. amyloliquefaciens* | + | + | + | ++ | + | + | + | + | ++ | ++ | + | – | + | ++ | + | + | – | + | – | + | + | + | + |
| #11 *B. amyloliquefaciens* | – | – | – | – | – | – | – | – | – | – | – | – | – | – | – | – | – | – | + | – | – | – | – |
| #13 *B. amyloliquefaciens* | – | – | – | ++ | – | + | + | + | ++ | ++ | – | – | – | – | – | ++ | ++ | – | + | – | + | – | ++ |
| #15 *B. amyloliquefaciens* | – | – | – | – | – | – | – | – | – | – | – | – | – | – | – | – | – | – | + | – | – | – | – |
| #07 *B. pacificus* | – | – | – | – | – | – | – | – | – | – | – | – | – | – | – | – | – | – | + | – | – | – | – |
| #09 *B. pacificus* | + | + | + | + | + | – | + | + | + | + | + | + | + | + | + | – | + | + | + | + | + | + | + |
| #03 *B. licheniformis* | + | + | + | + | ++ | ++ | ++ | ++ | ++ | ++ | + | + | + | ++ | + | + | + | – | + | + | ++ | ++ | + |
| #10 *B. safensis* | – | – | – | – | – | – | – | – | – | + | + | – | + | – | – | – | – | – | – | – | + | + | + |
| #01 *B. halotolerans* | + | + | – | + | + | + | + | + | + | + | + | + | + | + | + | + | + | ++ | + | – | – | + | + |
| #19 *B. halotolerans* | + | + | – | + | ++ | + | + | ++ | + | + | + | + | + | + | ++ | + | + | ++ | + | – | – | – | + |
| #21 *B. halotolerans* | – | – | – | + | – | – | + | + | + | + | – | – | – | – | + | + | + | – | – | – | – | – | – |
| #23 *P. polymyxa* | ++ | ++ | ++ | ++ | ++ | ++ | ++ | ++ | ++ | ++ | ++ | ++ | ++ | ++ | ++ | ++ | ++ | ++ | ++ | ++ | ++ | ++ | – |

**Notes.**
+/++ zone of inhibition observed with clear halo of growth inhibition; − no inhibition. The species designations were assigned based on the 16S rRNA gene analysis in Table 3.

**Table 7** Characterization of short-listed bacterial isolates.

| Strains # | #01 | #02 | #03 | #04 | #05 | #06 | #07 | #08 | #09 | #10 | #11 | #12 | #13 | #14 | #15 | #16 | #17 | #18 | #19 | #20 | #21 | #22 | #23 |
|---|---|---|---|---|---|---|---|---|---|---|---|---|---|---|---|---|---|---|---|---|---|---|---|
| **Salinity tolerance (NaCl)** | | | | | | | | | | | | | | | | | | | | | | | |
| 3%–5% | ++ | ++ | ++ | ++ | ++ | ++ | ++ | ++ | ++ | ++ | ++ | ++ | ++ | ++ | ++ | ++ | ++ | ++ | ++ | ++ | ++ | ++ | ++ |
| 6%–7% | ++ | ++ | ++ | ++ | ++ | ++ | ++ | ++ | ++ | ++ | ++ | ++ | ++ | ++ | ++ | ++ | ++ | ++ | ++ | ++ | ++ | ++ | – |
| 8% | ++ | ++ | + | ++ | ++ | + | – | + | ++ | ++ | ++ | ++ | ++ | ++ | ++ | ++ | ++ | ++ | ++ | ++ | ++ | + | – |
| 9% | + | ++ | – | ++ | ++ | – | – | – | ++ | ++ | ++ | ++ | ++ | ++ | ++ | ++ | ++ | ++ | ++ | ++ | ++ | – | – |
| 10% | – | + | – | + | + | – | – | – | + | + | + | + | + | + | + | + | + | + | ++ | + | + | – | – |
| 12% | – | – | – | – | – | – | – | – | – | – | – | – | – | – | – | – | – | – | – | – | – | – | – |
| **Bile salt tolerance (mole/L)** | | | | | | | | | | | | | | | | | | | | | | | |
| 0.1M -0.7M | – | – | – | – | – | – | – | – | – | – | – | – | – | – | – | – | – | – | – | – | – | – | – |
| **Antibiotic susceptibility** | | | | | | | | | | | | | | | | | | | | | | | |
| Kanamycin | – | – | – | – | – | – | – | – | – | – | – | – | – | – | – | – | – | – | – | – | – | – | – |
| Ampicillin | + | + | + | – | + | + | + | + | + | – | – | – | – | + | + | + | + | – | – | – | + | + | – |
| Tetracycline | – | – | – | – | – | – | – | – | – | – | – | – | – | – | – | – | – | – | – | – | – | – | – |
| Nalidixic acid | – | – | – | – | – | – | – | – | – | – | – | – | – | – | – | – | – | – | – | – | – | – | – |
| **pH tolerance** | | | | | | | | | | | | | | | | | | | | | | | |
| 5 | ++ | ++ | + | + | + | + | + | + | + | + | + | + | + | + | + | + | ++ | ++ | ++ | ++ | ++ | ++ | + |
| 6, 7, 8, 9 | ++ | ++ | ++ | ++ | ++ | ++ | ++ | ++ | ++ | ++ | ++ | ++ | ++ | ++ | ++ | ++ | ++ | ++ | ++ | ++ | ++ | ++ | ++ |
| **Thermal tolerance** | | | | | | | | | | | | | | | | | | | | | | | |
| 30 °C; 40 °C; 50 °C | ++ | ++ | ++ | ++ | ++ | ++ | ++ | ++ | ++ | ++ | ++ | ++ | ++ | ++ | ++ | ++ | ++ | ++ | ++ | ++ | ++ | ++ | ++ |
| **Oxygen requirement for growth** | | | | | | | | | | | | | | | | | | | | | | | |
| Aerobic | ++ | ++ | ++ | ++ | ++ | ++ | ++ | ++ | ++ | ++ | ++ | ++ | ++ | ++ | ++ | ++ | ++ | ++ | ++ | ++ | ++ | ++ | ++ |
| Microaerophillic | ++ | ++ | ++ | ++ | ++ | ++ | ++ | ++ | ++ | ++ | ++ | ++ | ++ | ++ | ++ | ++ | ++ | ++ | ++ | ++ | ++ | ++ | ++ |
| Anaerobic | – | – | – | – | – | – | – | – | – | – | – | – | – | – | – | – | – | – | – | – | – | – | – |
| **Enzyme production** | | | | | | | | | | | | | | | | | | | | | | | |
| Protease | +++ | + | – | + | – | +++ | + | ++ | ++ | + | + | + | ++ | – | + | + | ++ | ++ | +++ | – | ++ | + | ++ |
| Cellulase | ++ | ++ | ++ | ++ | ++ | ++ | + | ++ | + | – | ++ | ++ | ++ | ++ | ++ | ++ | ++ | ++ | – | – | ++ | ++ |
| Amylase | + | + | – | + | + | + | + | + | + | – | + | + | + | + | + | + | + | + | + | + | + | + | + |

**Notes.**

+ ability to grow under conditions based on colon formation; ++ and +++ increased size of colonies on plates; and - absence of growth. For enzyme production the symbols indicate the relative sizes of the zones of activity.

structurally diverse bacteriocins. Of the various types of marine samples collected, the antimicrobial producing isolates were most frequently recovered from sponges, followed by sediments, seaweeds, and sea-water samples. Marine sponges have multi-porous structures that may trap and maintain high bacterial densities, leading to higher recovery rates of antimicrobial producers. The recovery rate in marine sponges, at 7.6%, was at the lower end of the range reported in previous studies (5.5% to 50.0%) (*Laport & Muricy, 2008*). A lower recovery rate of antimicrobial producing isolates was also seen from seaweeds, at 3.6%, whereas previous studies had identified them from seaweeds at 11.0%–16.0% (*Lemos, Toranzo & Barja, 1985*; *Penesyan et al., 2009*). The differences between this study and previous studies could be due to geographical differences, marine conditions, or heat treatment to select the spore-formers that may eliminate the metabolically active vegetative cells.

The methods used to detect antimicrobial activity had different levels of sensitivity. The well-diffusion assay was less sensitive than the cross-streak assay. This effect was most obvious with the failure to detect activity against Gram-negative bacteria, *S. aureus* and MRSA, when the latter assay was used. This may occur because of the production of multiple antimicrobial compounds, with variable relative expression levels in liquid and solid media-based cultivation. Many studies have reported the influence of various factors on bacteriocin production in liquid culture such as, type of culture media; pH, temperature, growth phase, and quorum sensing regulation (*Gutowski-Eckel et al., 1994*; *Shanker & Federle, 2017*; *Yang et al., 2018*).

The most commonly identified species amongst the 23 shortlisted marine isolates that exhibited the strongest antimicrobial activities, were members of the *Bacillus subtilis* group. These species have been widely reported in both marine environments and terrestrial environments and are able to tolerate the broad environmental conditions (nutrients, pHs, chemo-physical conditions) that are typically found in marine environments. For example, a study by Ivanova *et al.*, reported that 55.0% (11/ 20) of endospore-forming bacteria isolated from different areas of the Pacific Ocean were *B. subtilis* species (*Ivanova et al., 1999*). *B. subtilis, B. amyloliquefaciens, B. pumilus* and *P. polymyxa* were all identified amongst aerobic spore-forming isolates from marine sources from the Gulf of Mexico, or isolated from seaweed samples collected from the Irish Sea (*Siefert et al., 2000*; *Luz Prieto et al., 2012*). Other typical marine species, for example, *B. aquimaris*, *B. algicola* and *B. hwajinpoensis* may require additional special nutrients or salts to recover or could have been eliminated during screening and shortlisting based on the strength of antimicrobial activity.

The marine *Bacillus/Paenibacillus* isolates that we have characterized had broad antimicrobial activity against a range of human, veterinary and food borne pathogens, including three antibiotic resistant pathogens (MRKP, MRSA and VRE). *B. amyloliquefaciens* #06, #08, #11, #13, *B. halotolerans* #01, #19, *B. licheniformis* #03, *B. safensis* #10, *P. polymyxa* #23 had broad antimicrobial activity against both Gram-positive and Gram-negative pathogen indicator strains and against other marine *Bacillus* of different species. This indicated production of either multiple antimicrobial compounds by a single *Bacillus,* or a broad-spectrum antimicrobial compound. Members of the *Bacillus* genus are known to produce

various types of antimicrobial compounds including polyketides, lipopeptides, bacteriocins, bacilysin, and volatile compounds (*Mondol, Shin & Islam, 2013*).The synergistic effects of these antimicrobials could result in a broad spectrum of antimicrobial activity, such as noted for a number of the isolates in this study. Also, production of broad-spectrum bacteriocin was recently reported for a marine *Bacillus*; sonorensin, identified from a marine *B. sonorensis* isolate, exhibited broad-spectrum antibacterial activity towards both Gram-positive and Gram-negative bacteria (*Chopra et al., 2014*). Amongst Gram-positive bacteria, such as *Bacillus* spp., the expression of bacteriocins and other antimicrobial compounds with activity against other Gram-positive bacteria is widespread and extensively studied. However, the production of compounds with activity against Gram-negative bacteria is less common, therefore, the isolates that have antimicrobial activity against Gram-negative bacteria are of particular interest. Rarely observed antimicrobial activities such as that observed against *Campylobacter, P. aeruginosa,* and even multidrug-resistant *K. pneumonia*, represent potentially novel compounds that, in future work, should be purified and structurally characterized. The *P. polymyxa* #23 isolate had the strongest and broadest activity and it is likely that the activity against Gram-negative bacteria results from the expression of polymyxin, which has long been known to have such activity (*Stansly & Schlosser, 1947*; *Poirel, Jayol & Nordmann, 2017*). The other isolates with activity against some of the Gram-negative bacteria tested, for example *B. amyloliquefaciens* #11, which had significant activity against both *Campylobacter* species but lesser activity against *E. coli* and *Salmonella*, appears to indicate a spectrum of activity that has not previously been reported and so the compound responsible may be novel and hence warrants further investigation.

The finding that many of the compounds that had activity against *C. perfringens* were inactivated by proteases, indicated that the antimicrobial compounds produced probably included bacteriocins. Interestingly, the production of large quantities of these antimicrobial compounds for drug development in the future could likely be achieved as it was demonstrated that most of the isolates were well adapted to a broad range of growth conditions (variations in nutrients, pH, salt concentration, and temperature). These characteristics of the isolates represent advantages that could facilitate manufacturing processes, product storage, and the potential harnessing of these isolates for in vivo use in animal or food applications.

These marine derived *Bacillus/Paenibacillus* isolates were also shown to have proteolytic, cellulolytic and amylolytic activity and hence may represent a promising source of important industrial enzymes such as proteases, cellulases, and amylases. Marine derived enzymes have been noted to have significant advantages in manufacturing because they commonly have high adaptability to high-salt concentration, and fluctuating temperature, pH, organic solvents, and ions (*Debashish et al., 2005*)

In conclusion, the bacteria in this collection of marine *Bacillus* isolates express a range of antimicrobial activities, some of which may represent novel compounds that warrant further study.

## CONCLUSIONS

It was hypothesized that the marine environment, particularly understudied regions abundant in varied marine habitats, such as the Vietnam Sea, would provide a rich source of bacteria that produce antimicrobial compounds. A survey of heat-resistant spore-forming bacteria found that 16.4% of isolates produced detectable levels of antimicrobial activity. Bacterial isolates were identified that had broad spectra of activity against both Gram-positive and Gram-negative pathogenic bacteria. Further analysis of a select group of isolates with the broadest activity profile showed that most of the antimicrobial compounds were sensitive to proteases, indicating that they were proteins rather than secondary metabolites. The study demonstrated that marine bacteria derived from the Vietnam Sea represent an interesting resource, producing antimicrobial compounds with activity against a range of clinically relevant bacterial pathogens, including important antibiotic resistant pathogens. Further biochemical characterization now needs to be undertaken to characterize the antimicrobial compounds, especially to define those that are novel.

### Funding

This work was financially supported by the Vietnam Academy of Science and Technology and an RMIT University PhD scholarship. The funders had no role in study design, data collection and analysis, decision to publish, or preparation of the manuscript.

### Grant Disclosures

The following grant information was disclosed by the authors:
Vietnam Academy of Science and Technology.
RMIT University PhD scholarship.

### Competing Interests

The authors declare there are no competing interests.

### Author Contributions

- Khanh Minh Chau conceived and designed the experiments, performed the experiments, analyzed the data, prepared figures and/or tables, authored or reviewed drafts of the paper, and approved the final draft.
- Dong Van Quyen and Andrew T. Smith conceived and designed the experiments, authored or reviewed drafts of the paper, and approved the final draft.
- Joshua M. Fraser performed the experiments, authored or reviewed drafts of the paper, and approved the final draft.
- Thi Thu Hao Van conceived and designed the experiments, performed the experiments, authored or reviewed drafts of the paper, and approved the final draft.
- Robert J. Moore conceived and designed the experiments, analyzed the data, prepared figures and/or tables, authored or reviewed drafts of the paper, and approved the final draft.

## Field Study Permissions

The following information was supplied relating to field study approvals (i.e., approving body and any reference numbers):

Field collection of samples was approved by the Nha Trang Institute of Technology Research and Application.

## DNA Deposition

The following information was supplied regarding the deposition of DNA sequences:

The 16S rRNA gene sequences are available at GenBank: MT758446 to MT758468.

## Data Availability

The data is available in Tables 2–7 and Fig. 3.

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
