# Peer review of "Broad spectrum antimicrobial activities from spore-forming bacteria isolated from the Vietnam Sea"

_PeerJ, doi:10.7717/peerj.10117_

## Round 0.1 · original submission · Major Revisions

Please take into consideration the reviewer’s comments and provide back a point-by-point rebuttal letter addressing those concerns. In particular, please show the relevance and timeliness of your research since it is questioned its novelty by one of the reviewers. More in-depth experimental evidence of novel findings is also critical for further consideration.

·

Basic reporting

This manuscript presents relevant information about the use of spore-forming bacteria to inhibit pathogenic bacteria growth. However, some sections of the presented data can be improved. For this reason, I considered that this manuscript needs minor changes for being considered for its publication in this journal.

Experimental design

Include more detail in enzymatic assays protocol performed in this manuscript.
Include more detail in the experimental design, highlight statistical factors and variables of response in the statistical analyses applied to the findings of this research.

Validity of the findings

Include a possible mode of action of spore-forming strains against pathogenic bacteria tested.
Try to compare the obtained findings with similar assays were spore-forming bacteria were applied to inhibit pathogenic bacteria growth.

Additional comments

Dear Author, I reviewed the manuscript (51002v1) entitled Broad spectrum antimicrobial activities from spore-forming bacteria isolated from the Vietnam Sea. This manuscript presents relevant information about the use of spore-forming bacteria to inhibit pathogenic bacteria growth. However, some sections of the presented data can be improved. For this reason, I considered that this manuscript needs minor changes for being considered for its publication in this journal.

Additional comments.
Highlight the advantages of using nowadays these spore-forming strains against pathogenic bacteria.
Check paragraphs extension in this manuscript.
Include more detail in enzymatic assays protocol performed in this manuscript.
Include more detail in the experimental design, highlight statistical factors and variables of response in the analyses applied to the findings of this research.
Include a possible mode of action of spore-forming strains against pathogenic bacteria tested.
Try to compare the obtained findings with similar assays were spore-forming bacteria were applied to inhibit pathogenic bacteria growth.
Include future trends to keep working with the obtained data.
Try to conclude with a general statement of the most relevant part of this study.

Reviewer 2 ·

Basic reporting

The article needs to improve, it presents an insufficient introduction and a background is missing to demonstrate how the work fits into the broader field of knowledge. Current references are missing.
Contaminated culture in figure 1; Low quality of figure 3; Figure 4 is a catastrophe, the phylogenetic tree must be rebuilt following all the required standards.
There are many tables, some unnecessary and incomplete essential information.

Experimental design

The results are very preliminaries. The information is not new in relation to marine bacteria, mainly for the genus Bacillus. Additional experiments are needed for the study, such as isolation and identification of bioactive compounds, tests for minimum inhibitory concentration and minimum bactericidal concentration, and cytotoxicity tests.
The reference list has many errors, this demonstrates a lack of care in preparing the document.

Validity of the findings

This study does not bring news in the area, it is a weak repetition of other publications.
In view of its lack of new information, I suggest that the manuscript be rejected.

Additional comments

The study would be interesting if more results were presented. The work must be continued in order to isolate the bioactive compound and show that it has some potential to be used in the treatment of any infection.

---

## Round 0.2 · accepted · Accept

Thanks for addressing the revisions requested. Now your manuscript is accepted in PeerJ.

·

Basic reporting

This version of the manuscript followed all the suggested modifications and recommendations by the reviewers.

Experimental design

This section was improved in the revised version of the manuscript.

Validity of the findings

The obtained findings in this research are well described and compared with bibliographical references and justify the importance of this obtained data.

Additional comments

Dear Author, I reviewed the revised version of the manuscript (51002v2) entitled: Broad spectrum antimicrobial activities from spore-forming bacteria isolated from the Vietnam Sea. This version of the manuscript followed all the suggested modifications and recommendations by the reviewers. Besides, the findings obtained in this research are well described and compared with bibliographical references and justify the importance of this obtained data. For this reason, I considered that this manuscript can be accepted for its publication in this journal.